# DABS 2.0: Improved Datasets and Algorithms for Universal Self Supervision

**Alex Tamkin**[†]
Stanford University

**Gaurab Banerjee**
Stanford University

**Mohamed Owda**
Stanford University

**Vincent Liu**
Stanford University

**Shashank Rammoorthy**
Stanford University

**Noah D. Goodman**
Stanford University

## Abstract

Universal self-supervised learning (SSL) algorithms hold enormous promise for making machine learning accessible to high-impact domains such as protein biology, manufacturing, and genomics. We present DABS 2.0: a set of improved datasets and algorithms for advancing research on universal SSL. We extend the recently-introduced DABS benchmark with the addition of five real-world science and engineering domains: protein biology, bacterial genomics, multispectral satellite imagery, semiconductor wafers, and particle physics, bringing the total number of domains in the benchmark to twelve. We also propose a new universal SSL algorithm, Capri, and a generalized version of masked autoencoding, and apply both on all twelve domains—the most wide-ranging exploration of SSL yet. We find that multiple algorithms show gains across different domains, outperforming previous baselines. In addition, we demonstrate the usefulness of DABS for scientific study of SSL by investigating the optimal corruption rate for each algorithm, showing that the best setting varies based on the domain. Code will be released at http://github.com/alextamkin/dabs.

## 1 Introduction

Recent months have continued to see the rise of large, self-supervised learning (SSL) models across multiple domains [28, 31, 14, 4, 21, 77]. These works have been characterized by an increasing convergence upon a similar set of methods [71, 10] generally involving large transformer model architectures trained on large-scale datasets. Despite this trend, the actual learning tasks used to train these SSL models still tend to vary significantly: contrastive, autoregressive, and denoising objectives have each claimed their own niche, and different techniques still predominate in different communities. While some prior work has moved towards a more domain-agnostic approach to SSL, these works have largely been limited to analyzing the well-studied domains of images, text, and speech [3, 6], leaving open the question of generalization to less-studied domains, including scientific, medical, and engineering settings.

The DABS benchmark [62] was developed to provide a testbed for research on universal SSL algorithms that could be applied across seven different domains, including less common settings such as wearable sensors and chest x-rays. DABS can be used to develop new and improved universal SSL algorithms, or to conduct scientific studies on pretraining and transfer across diverse domains. Crucially, **evaluating on a breadth of different domains** enables researchers to have greater confidence that their methods will generalize to a range of different settings in the real world, and also to investigate how design choices made for one domain can affect learning on others.

---

[†]`atamkin@stanford.edu`

36th Conference on Neural Information Processing Systems (NeurIPS 2022) Track on Datasets and Benchmarks.

While one goal of the DABS benchmark is to support research for underserved domains where there is less research on pretraining, three of seven DABS domains in the original paper were English text, natural images, and speech recordings, which have already received ample attention as settings for self-supervised learning. To accelerate research in high-impact domains where data is prevalent but human labels are scarce, we introduce **DABS 2.0, adding five new science and engineering domains** to the existing 7 domains in the DABS benchmark, each containing their own pretraining and transfer datasets. Importantly, the datasets in these domains were curated and created with the help of domain experts, and center on real-world tasks such as detecting defective semiconductor wafers or identifying exotic particles. The addition of these domains enable us to conduct the widest-ranging study of self-supervised learning yet.

We also introduce **two new universal SSL algorithms** and evaluate them on all twelve domains. The first is a generalized version of masked sequence modeling, also referred to as masked autoencoding (MAE), an approach that has seen success when applied to text [17], images [28], and videos [21, 66]. The second is a contrastive-masked algorithm called Capri that generalizes approaches previously explored in natural images [67] and audio [78], and relaxes some modality-specific components required by MAE.

Finally, we demonstrate the usefulness of DABS for studying the science of self-supervised learning by evaluating three algorithms on all 12 domains across **three different corruption fractions**, controlling the difficulty of the self-supervised task (i.e. what fraction of embeddings are masked or permuted). The resulting methods show considerable gains on certain domains. However, this improvement is not uniform across domains, revealing an important direction for future work. We also contribute new functionality to the DABS codebase, enabling **easy execution** of pretraining and transfer runs in sequence on a given accelerator to facilitate easy experimentation.

We hope these contributions help advance the study of universal self-supervision, enabling better scientific understanding and practical advancement of SSL, resulting in positive impact on real-world problems.

## 2 Domains and Datasets

Here, we describe the new datasets in DABS 2.0. In the original DABS paper [62], the benchmark domains represent a range of research communities, including communities with large bodies of work on self-supervised learning (e.g. text, natural images) to domains with more nascent streams of research (medical imaging, sensor recordings). In DABS 2.0, we bolster our focus on the latter group by adding five domains representing science and engineering fields. Importantly, the datasets in all five domains were created or curated with the help of domain experts. As in the original DABS paper [62], we choose open-access datasets, in particular preferring datasets that could be automatically downloaded given the large number of datasets in the benchmark (57). Some examples of the pretraining datasets from each domain are depicted in Figure 1, left. Similar to the original DABS domains, dataloading and preprocessing within each dataset has been standardized to ensure fair comparisons; more information about data processing for each domain is provided in the Appendix.

**Bacterial Genomics** Genomic sequences are similar to text domains in that both contain sequences of discrete tokens. With new species of bacteria being discovered and sequenced every year, the field of bacterial genomics is not only data rich, but also offers the opportunity to explore how self-supervised methods generalize under temporal distributional shifts. We pretrain using the training set of the Genomics OOD Dataset [53], consisting of 1M DNA sequences across 10 bacterial classes discovered before 2011. We evaluate transfer on the in-distribution validation set of the Genomics OOD dataset, containing 100,000 labeled examples from those same bacterial classes, and on the out-of-distribution validation set containing 600,000 examples across 60 bacterial classes discovered between 2011 and 2016. To prepare the input for models, we tokenize each genomic sequence at the nucleobase level: adenine (A), cytosine (C), guanine (G), and thymine (T).

**Semiconductor Wafer Manufacturing** While natural images are a popular domain for applying SSL techniques, it remains unclear whether natural image-centric strategies will generalize to industrial settings, such as detecting defects in semiconductor wafers. To assess how SSL techniques perform on such real-world images we consider the WM-811K[74] dataset, a corpus of semiconductor wafer measurements labeled with their specific class of defect (e.g. edge-ring, donut, center, local,

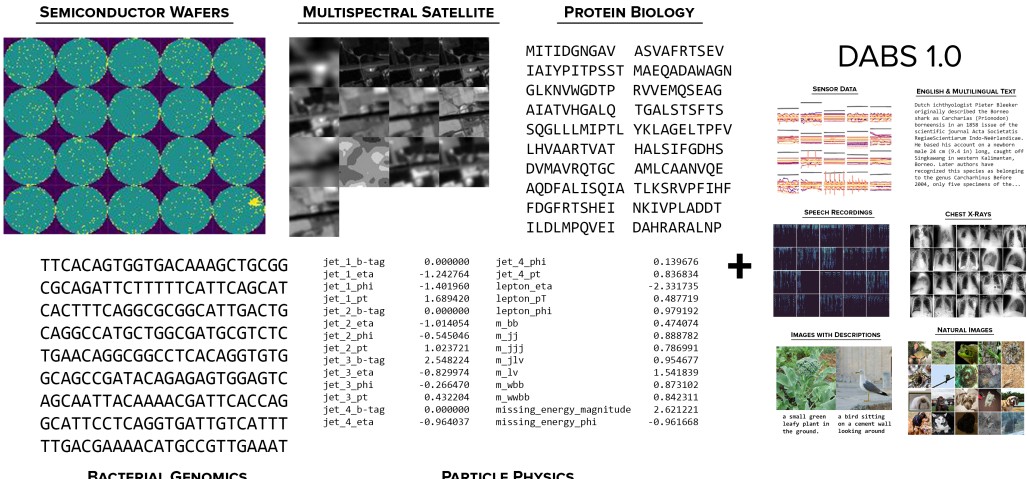

Figure 1: **Left: The five new domains in DABS 2.0.** From left to right, then top to bottom: Semiconductor wafers, multispectral satellite imagery, protein biology, bacterial genomics, particle physics. Examples taken from the pretraining dataset of each domain. 20 wafer examples are shown for semiconductors, a single input is shown for all other examples. Satellite imagery shows all 13 channels of a single multispectral image. See Section 2 for more information about each of the new domains. **Right: The seven original domains from DABS 1.0** that we also train and evaluate on. From left to right, then top to bottom: Wearable Sensors, English Text and Multilingual Text, Speech Recordings, Chest Xrays, Paried Image + Text, Natural Images.

scratch) or lack of defect. We pretrain on the 638,597 unlabeled examples from the WM-811K dataset. For transfer accuracy, we evaluate on the remaining 172,950 labeled wafers from WM-811K. The data for each wafer map is a 2D array of scalars where 0 represents the background of the wafer, 1 represents dice (semiconducting materials) that are not defective, and 2 represents dice that are defective [74]. To prepare the input for the model, we convert the 0,1,2 representation into grayscale pixels (black, 50% grey, and white) to produce a two-dimensional image.

**Particle Physics** High energy physics, also known as particle physics, is one of a growing number of scientific communities using deep learning to gather insights from their data. These datasets are often tabular in nature, and have the potential to contain millions of examples due to the large-scale nature of these experiments. We pretrain on a randomly selected 9.9M instances from the HIGGS dataset [7], a particle physics benchmark containing 21 kinematic properties, and 7 functions of these kinematic properties, for 11M Monte Carlo simulations of particle collisions. The task is to distinguish between particle collision simulations generated by a *signal process*, involving a Higgs boson, from a *background process* that produces the same resulting particles but that does not involve a Higgs boson. We evaluate transfer on the remaining 1.1M instances of the HIGGS dataset.

**Protein Biology** Protein databases have increased exponentially in size over the past several years [15], with much of this data lacking additional human annotations. At the same time, recent advances have shown SSL to be a powerful tool for extracting knowledge from unlabeled protein sequences [50]. To determine the success of domain-agnostic approaches with respect to learning from protein sequences, we pretrain on Pfam [20], a database with 31M protein sequences used commonly in bioinformatics research. We evaluate transfer on several tasks from the TAPE benchmark [50], namely: the Fluorescence [56] dataset, Remote Homology Detection dataset (using the training and validation sets from [36], derived from the SCOP 1.75 database [23] [42]), Secondary Structure Prediction dataset (training and validation sets from [36], with data derived from the Protein DataBank [9]), and the Stability [54] dataset. We tokenize the protein sequences at the amino-acid level to create inputs for the model. These tasks and associated datasets are described in further detail in the TAPE benchmark [50].

**Multispectral Satellite Imagery**    While similar in some respects to natural images, satellite imagery differs in that remote sensing instruments may often capture a range of spectral bands beyond the typical RGB colorspaces, including near infrared and shortwave infrared. These additional spectral bands may be useful for a range of environmental or social planning purposes by providing information about land temperatures, soil water content, or correcting for atmospheric effects like clouds or precipitation [29]. However, these bands often have different characteristics from typical RGB images, making certain techniques (e.g. colorspace-based augmentations) inapplicable. To test how domain agnostic SSL algorithms perform on satellite imagery, we pretrain on a randomly selected 24,300 examples from EuroSAT [29], a dataset constructed from Sentinel-2 satellite images covering 13 spectral bands. We evaluate transfer on the remaining 2,700 examples from EuroSAT.

**Existing DABS 1.0 Domains**    These domains join the seven existing domains from the original DABS paper: Natural Images, Speech Recordings, English Text, Multilingual Text, Wearable Sensors, Chest X-Rays, Paired Images and Text (Figure 1, right), bringing the total number of domains in the benchmark to twelve. See the original DABS paper [62] for more information about each of these domains. In the following sections we train models for all twelve domains.

## 3    Algorithms

The original DABS paper [62] presented the first universal SSL algorithms evaluated across seven different pretraining datasets. Here, we present two additional domain-agnostic algorithms and evaluate their performance on the new full suite of 12 DABS domains. For all algorithms, we leverage the same Domain Agnostic Transformer approach used in the original DABS paper [62], which uses a set of embedding modules to map inputs to sequences of embeddings (e.g. via token, patch, or segment embeddings) then concatenates them as input to an encoder-only transformer model.

### 3.1    Generalized Masked Autoencoding

One popular strategy for self-supervised learning is masked sequence modeling, also known as masked autoencoding (MAE). A breakthrough instantiation of this method was BERT [17] in the context of natural language text (although the roots of the idea extend much earlier [70, 45]), and it has recently seen a new wave of adaptation for continuous domains such as audio, images, and video [28, 25, 66, 21].

We generalize the MAE framework to train on all 12 DABS domains as follows: For **tokenized domains** (English and multilingual text, bacterial genomics, and proteins) we predict the missing token with a softmax layer and use the negative log-likelihood loss. For **continuous domains** (natural images, chest x-rays, speech, sensors, semiconductor wafers, particle physics, multispectral satellite imagery) we directly predict the output token and apply a mean-squared error loss. For **multimodal** datasets (image and text) we apply the respective loss to each tokenized or continuous modality within the input.

While this generalization is a straightforward way to evaluate MAE across the DABS domains, we note that it is not without complications. First, the MAE paradigm is somewhat less general than other universal SSL methods because the loss is dependent on the domain of the input (or part of the input). Furthermore, prior work has identified several design choices whose optimal settings appear to differ across tokenized and continuous domains, namely: 1) leveraging additional decoder layers has been shown to be helpful for continuous data, as has 2) entirely dropping the masked tokens from the attention computation [28, 21]. Finally, in this work we consider only linear evaluation, while masked sequence models have typically shown greater performance than competing methods when finetuned.

These complexities aside, the strength and generality of MAE merits its study as a domain-agnostic method, and future work can explore the relative tradeoffs of each of these design choices in a domain-agnostic context.

### 3.2    Capri: A Hybrid Masked-Contrastive Algorithm

Given the slight additional complexities of the MAE paradigm, we explore an algorithm that attempts to blend the strengths of contrastive learning [27, 19] and MAE, and which does not require a different

|  |  | Cap | Gen | Med | Nat | Par | Pro | Sat | Sem | Sen | Spe | Tex | Mul |
|---|---|---|---|---|---|---|---|---|---|---|---|---|---|
| None |  | 50.1 | 22.4 | 68.1 | 10.1 | 54.8 | 29.9 | 62.3 | 77.7 | 69.8 | 24.9 | 42.3 | 58.1 |
| ShED | 15% | **52.3** | 25.6 | 69.8 | 16.1 | 68.0 | **36.8** | 57.8 | 92.4 | 85.2 | **42.1** | **46.6** | 57.5 |
|  | 50% | 50.7 | 17.1 | 70.6 | 19.4 | 67.2 | 35.9 | 55.2 | 92.3 | 65.2 | 29.1 | 45.3 | **64.0** |
|  | 85% | 51.5 | 19.8 | **73.2** | **24.6** | 60.3 | 29.9 | 61.5 | 91.4 | 85.0 | 37.6 | 44.3 | 48.2 |
| Capri | 15% | 51.6 | 19.2 | 70.0 | 21.0 | DV | 23.5 | 11.1 | 91.2 | DV | 28.9 | 42.1 | 57.6 |
|  | 50% | 50.7 | 22.6 | 70.4 | 19.6 | DV | 19.5 | 67.4 | 91.8 | DV | 22.3 | 42.8 | 57.6 |
|  | 85% | 51.4 | 16.7 | 52.4 | 21.3 | DV | 18.0 | 63.3 | 92.5 | DV | 21.7 | 40.2 | 57.5 |
| MAE | 15% | 51.4 | 26.6 | 71.3 | 19.8 | 68.7 | 32.2 | 84.1 | 93.0 | **85.3** | 25.6 | 44.3 | OM |
|  | 50% | 50.2 | **39.0** | 70.8 | 19.4 | **70.0** | 30.9 | **86.3** | 92.9 | 82.5 | 27.2 | 43.9 | OM |
|  | 85% | 50.0 | 25.7 | 70.6 | 22.4 | 63.5 | 24.2 | 84.8 | **93.9** | 77.6 | 29.4 | OM | OM |

Table 1: Average of transfer metrics for different corruption fractions. Runs marked with "DV" indicate cases where training diverged. "OM" indicates cases where the large vocabulary size of the model produced an an out-of-memory error for the given batch size. **Legend:** Cap: Captioned Images, Gen: Genomics, Med: Medical Images, Nat: Natural Images, Par: Particle Physics, Pro: Protein Biology, Sat: Satellite Images, Sem: Semiconductor Wafers, Sen: Wearable Sensors, Spe: Speech Recordings, Tex: English Text, Mul: Multilingual Text

output format per modality. Intuitively, the algorithm predicts the embeddings of a masked sequence with a contrastive loss, using the other embeddings in the sequence as negative examples. We call this algorithm *contrastive prediction of redacted embeddings*, or **Capri** for short.

Concretely, given a set of input tokens $x = \{x_0, \ldots, x_k\}$ as input to the transformer, we create a masked $\tilde{x} = \{\tilde{x}_0, \ldots, \tilde{x}_k\}$ where $\tilde{x}_i = x_i$ with probability $1 - p$ and equals the zero vector $\vec{0}$ otherwise. The transformer then generates predicted embeddings $\hat{x} = \{\hat{x}_0, \ldots, \hat{x}_k\}$ and the loss for each masked token $x_i$ is computed as:

$$\mathcal{L}(x_i) = \frac{\exp\left(\text{cosine-similarity}(\hat{x}_i, x_i)/\tau\right)}{\sum_j \exp\left(\text{cosine-similarity}(\hat{x}_i, x_j)/\tau\right)} \tag{1}$$

where cosine-similarity$(x, y) = \frac{x}{||x||_2} \cdot \frac{y}{||y||_2}$, the dot product of two normalized vectors, and $\tau$ (set to 0.07 in our experiments) controls the temperature of the softmax.

Capri can be seen as a bidirectional masked variant of contrastive predictive coding [68], and instantiations of this approach have been applied to specific modalities such as vision [67] and audio [78]. We apply this SSL framework across the 12 DABS domains, including the multimodal text-image domain, exploring its approach as a domain-agnostic SSL method.

### 3.3 Shuffled Embedding Detection (ShED)

Finally, we also evaluate ShED [62], a shuffled embedding detection algorithm which permutes a subset of the embeddings for an input (prior to adding position embeddings) and trains a classifier to predict which embeddings were perturbed. See [62] for more details about ShED. For simplicity we do not consider the eMix algorithm from the original DABS paper [62].

## 4 Investigating the Optimal Corruption Rate Across Algorithms and Domains

One similarity of each algorithm discussed in Section 3 is that each applies a corruption transformation to a fraction of the input embeddings: ShED permutes a fraction of the embeddings, while MAE and Capri mask out a given fraction. The choice of this fraction (which we will term the *corruption rate*) determines the difficulty of the self-supervised task. If the corruption rate is too small, the task will be too easy and the model will learn slowly. Too large, and the model may not be able to learn from the resulting example.

Several works have studied the impact of the corruption rate on MAE-type models, including in text [71], images [28], and videos [21]. The outcomes of these studies seem to suggest that the optimal masking rate is highly dependent on the domain: text for example appears to require a lower masking fraction than images and especially video.

One hypothesis for this diversity might be termed the *redundancy hypothesis* [28]: that domains vary in the amount of redundant information they contain across parts of the input. For example, images may have more redundant structure across spatial patches than text does across tokens—patches of sky tend to be near other patches of sky—while words are typically rarely repeated several times in a row in written text. Thus, inputs with more redundant structure require larger corruption rates to make the task challenging enough. However, it is difficult to answer these kinds of data-dependent questions when studying self-supervised learning across only $N = 3$ domains, each in a separate study with different experimental settings.

To demonstrate the utility of DABS for exploring these questions, we conduct a large-scale study of optimal corruption rates across all twelve domains and three algorithms. Despite requiring over 500 runs across 57 datasets, this study is simple to carry out with the DABS codebase, requiring only several commands of the following form which perform the requisite pretraining and transfer runs on the provided device:

```
python3 -m scripts.train_single_domain   \
    --domain=genomics                     \
    --algorithm=mae                       \
    --corruption_frac=0.5
```

We reuse the experimental settings and hyperparameters from the original DABS paper: We use a Transformer [69] with 12 layers, hidden size 256, 8 attention heads, and dropout with 0.1 probability. Inputs are mapped to a sequence of embeddings using a small set of embedding modules (patch/segment embeddings for continuous data, and token embeddings for tokenized data). We train 100k steps for pretraining and 100 epochs of linear evaluation transfer, where we train a linear classifier on top of the frozen pretrained model. We use the AdamW optimizer [40] with learning weight and weight decay both set to 1e-4. The one change we make from the original DABS paper is that we truncate long transfer runs at 100k steps, as several of the DABS 2.0 transfer datasets are quite large. See the original DABS paper [62] or the DABS codebase[2] for more thorough experimental settings and details.

## 5   Results

The results of these experiments are summarized in Table 1, which shows the average validation metric across transfer datasets for the given pretraining algorithm, domain, and corruption rate.

**Takeaways**   Overall across all 12 domains, we see at least one algorithm, and often two or all three, showing clear gains from pretraining. In particular, both MAE and ShED show gains from pretraining in almost all cases, demonstrating their promise as general SSL approaches. There are some clear trends within a domain: for example, MAE significantly outperforms the other methods on multispectral satellite imagery, while ShED proves superior on protein data. MAE also exhibits a limitation for tokenized datasets: large vocabulary sizes and larger masking fractions can cause out-of-memory error due to the cost of the softmax operation ("OM" in Table 1). Capri appears to generally perform worse than ShED and MAE, and its training diverges on certain datasets ("DV" in Table 1). Despite the strong performance of MAE and ShED, there does not appear to be a clear pattern that would enable choosing the optimal algorithm and corruption rate *a priori*. An automated way of determining this rate may be a promising direction for future work.

**Contextualizing the performance of MAE**   While MAE performs strongly in some domains (e.g. especially in satellite imagery), it is important to reemphasize differences in our experimental setup to previous work on MAE in text [17], image [28], and video [21] datasets. First, the benefits of MAE have been shown to be strongest in the finetuning setting, rather than the linear evaluation setting. Second, in continuous domains such as images and videos, MAE-trained models are improved upon with the use of additional decoder layers on top of the encoder backbone. Finally, in continuous settings, MAE-trained models have been shown to perform better (and are far more efficient) when the masked tokens are dropped from the transformer computation entirely, rather than merely masked out.

---

[2]http://github.com/alextamkin/dabs

# 6   Related work

Here we discuss several streams of work related to the contributions in DABS 2.0. See the original DABS paper [62] for a more comprehensive discussion of work on self-supervised learning (SSL) and domain-agnostic methods, Section 3 for discussions of prior work related to the algorithms explored in this work, and [8, 22, 10, 61] for broader perspectives on these trends.

**Self-supervised learning for science and engineering**   Several bodies of work attempt to apply self-supervised learning to science and engineering tasks. For example, works have applied SSL to proteins [51], RNA [12], organic molecules [55], wearable sensors [65], multispectral satellite imagery [41], semiconductor manufacturing [33], medical data [80, 60, 26] and high energy physics [18], among many others. These domains make for promising sites to apply SSL methods because many scientific instruments regularly produce large amounts of data, but it is often expensive to hire domain experts to annotate this data for supervised machine learning. The inclusion of the DABS 2.0 domains in the benchmark is intended to drive progress in generalizable SSL algorithms which could benefit all of these fields, including ones where good techniques for SSL are not yet known.

**Scientific investigations of pretraining and transfer**   Several works systematically study the various factors influencing pretraining or transfer, either as a way to improve the performance of models or solely to attain a better scientific understanding of the self-supervised learning process. For example, several works vary the pretraining data distribution [11], difficulty of the pretraining task [71], pretraining hyperparameters [39], or choice of pretraining algorithm [34]. Other works focus more on the interface between pretraining and transfer, exploring what kinds of dataset shifts influence the success of transfer [35, 16, 75], which parts of the network matter most for transfer [76, 48, 64], or how the choice of transfer method influences the accuracy [79, 72], efficiency [52, 58, 37] or robustness [73, 30, 63] of the resulting model.

# 7   Discussion

## 7.1   Experimental Scope and Limitations

The past year has seen vigorous discussion about the relationship between benchmarks in machine learning and their connection to real world goals and problems [57, 46, 49]. Here, we discuss several of these concerns in connection to the DABS 2.0 benchmark, especially given its focus on real-world science and engineering datasets curated by domain experts. We discuss several concerns through the lenses of *internal* and *external validity* [38]:

**Internal validity** concerns the experimental procedures conducted within a specific benchmark. DABS attempts to reduce as much possible experimental variation by providing an easy-to-use codebase with easy-to-modify baseline algorithms and standardized preprocessing of datasets. However, one unresolved challenge here is choosing good hyperparameters for our corruption fraction experiments. This issue has been shown to be subtle and challenging for a single algorithm applied to a single dataset within a domain [13], and only compounds when expanding to multiple domains and algorithms. In addition, groups with a larger budget for hyperparameter search may see larger gains for a given algorithm than a less well-resourced organization would, making it challenging to fairly compare algorithms.

**External validity** refers to the degree to which experimental insights have relevance to the rest of the real world. We discuss two major sub-challenges here: First, *construct validity* [44] asks whether a particular metric used in a research study corresponds to the actual task or behavior of interest. In particular, [49] question whether datasets that seek to measure "general" capabilities in a particular domain (e.g. language understanding or visual understanding) faithfully realize that goal. We agree that strong claims such as generality require strong evidence, and for this reason the DABS benchmark does not take the position that the datasets in each domain represent "general" capabilities. However, they do represent a range of possible downstream tasks in each domain, and can measure how methods might perform on similar tasks. While this task distribution may not cover the full space of possible tasks one might like to address, the DABS 2.0 domains were chosen because they were constructed by domain experts to target problems with real-world importance. Self-supervised

learning methods that help domain experts achieve better performance on tasks that are important to them may provide real-world value independent of any abstract claims of generality.

A second external validity concern is that of *ecological validity,* also referred to as *mundane realism* [24]. This notion refers to whether the results of a scientific study generalize to the real world. While we have discussed the individual DABS domains, the goal of the DABS benchmark is to understand the behavior of SSL across domains and produce algorithms that generalize to new domains. The DABS 2.0 domains were chosen because they reflect the settings that are especially promising for SSL—data-rich but label-scarce settings with significant potential for scientific impact. One potential challenge is that a user of the benchmark could attempt to hardcode an "if statement" of domain-specific algorithms in an attempt to game the system, but the solution would be unlikely to be adopted by the community as it would not generalize, and would fail when new domains (e.g. the DABS 2.0 tasks) are introduced to the benchmark. Another limitation is that 12 domains is still a small number compared to the vast array of domains in the world, and the DABS benchmark does not yet have coverage for several important modalities, such as point clouds and graphs. However, it is surely an improvement over studying two or three domains, as is common practice, and provides a template for continued expansion into new domains.

### 7.2 Societal Impact

It is challenging to forecast the impacts of domain-agnostic SSL due to the wide-ranging fields it could be applied to. In DABS 2.0, we aim to introduce science and engineering domains where advances in these fields could lead to the development of improved medicines, more affordable electronics, or sustainable development. By the same token, however, advances in each field—whether due to DABS or other sources of scientific progress—could enable malicious users to cause harm. Technology does not exist in a vacuum, and effective governance frameworks and professional norms are important to ensure positive outcomes from technological progress. In this work, we also aimed to model how DABS could be used for systematic evaluation and understanding of SSL algorithms across a range of possible pretraining and transfer datasets. A broad coverage of different domains could help users of the benchmark identify failure modes of existing systems. For more discussion of societal impacts of domain-agnostic SSL, see the original DABS paper [62].

### 7.3 Future Work

We see ample opportunity for future work with DABS. Most directly, DABS enables the testing and development of improved SSL algorithms that perform better across the 12 domains in the benchmark. Another important line of work is in scaling existing algorithms to larger models and compute budgets, to see how close existing algorithms fare to state-of-the-art models typically trained on far more data. Finally, we have demonstrated in this paper the utility of DABS for easily conducting experiments for various experimental parameters (e.g. the corruption fraction), drawing scientific insights about the benefits and tradeoffs of different SSL algorithms. Our exploration has only barely scratched the surface of this kind of analysis, which remains ripe for future study.

## 8 Conclusion

We introduce DABS 2.0, augmenting the DABS benchmark for universal self-supervision with 5 additional science and engineering domains, two new algorithms, and the widest exploration of SSL yet across 12 domains and different corruption fractions. We hope this demonstrates the utility of DABS for easily creating and evaluating new SSL algorithms for the real-world settings where they may have the most positive impact.

## Acknowledgments and Disclosure of Funding

We would like to thank Shyamal Buch and Alex Ku for useful discussions and feedback. AT is supported by an Open Phil AI Fellowship.

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
