# A  Dataset Licenses

Below we list each dataset's license, as provided either in the paper proposing the dataset or on the dataset website. For datasets where we were unable to find a license, we list "No License."

- **Bacterial Genomics:** Genomics OOD Dataset (Apache License, Version 2.0)
- **Semiconductor Wafer Manufacturing:** WM-811K (CC0: Public Domain)
- **Particle Physics:** HIGGS (No License)
- **Protein Biology:** Pfam (CC0: Public Domain), Fluorescence (No License), [32] (No License), SCOP 1.75 (CC-BY 4.0), [36] (No License), Protein DataBank (CC0: Public Domain), Stability (No License)
- **Multispectral Satellite Imagery:** BigEarthNet (Community Data License Agreement – Permissive, Version 1.0 [3]), EuroSat (CC0: Public Domain)

# B  Origins and Collection of Datasets in DABS 2.0

DABS makes use of a diverse array of kinds of data. Here, we detail to the best of our knowledge how these datasets were collected, including whether consent was explicitly obtained from humans providing the data.

- For WM-811K, Wu et. al [74] note "the [dataset] was built comprising 811,457 wafer maps, in which each wafer map was collected from real-world fabrication. Domain experts were recruited to annotate the pattern type for approximately 20% of the wafer maps in the WM-811K dataset."
- For HIGGS, Baldi et. al [7] state "simulated events are generated with the MADGRAPH [5] event generator assuming 8 TeV collisions of protons as at the latest run of the Large Hadron Collider, with showering and hadronization performed by PYTHIA [59] and detector response simulated by DELPHES [43]".
- For Genomics OOD, Rie et. al [53], "downloaded 11,672 bacteria genomes from National Center for Biotechnology Information (NCBI [4]) on September 2018 ... Genomes belonging to new classes that were first discovered between 01/01/2011 and 01/01/2016 are used for generating the validation dataset for OOD. Genomes belonging to the old classes but sequenced and released between 01/01/2011 and 01/01/2016 are used for generating the validation dataset for in-distribution. Similarly, genomes belonging to the new classes that were first discovered after 01/01/2016 are used for generating test dataset for OOD, while genomes belonging to the old classes that were sequenced and released after 01/01/2016 are used for generating the test dataset for in-distribution ... To mimic the real sequencing data, we fragmented genomes in each class into short sequences of length 250 base pairs, which is a common length that the current sequencing technology generates. Among all the short sequences, we randomly choose 100,000 sequences for each class for the training, validation, and test datasets."
- For the training and validation sets from the DeepSF paper, Hou et. al [32] state, "The main dataset that we used for training, validation and test was downloaded from the SCOP 1.75 genetic domain sequence subsets with less than 95% pairwise identity released in 2009 ... The dataset contains 16 712 proteins covering 7 major structural classes with total 1195 identified folds."
- For SCOP 1.75, Murzin et. al [42] say the dataset, "includes all proteins in the current version of the PDB [2] and almost all proteins for which structures have been published but whose co-ordinates are not available from the PDB."
- For Pfam, the data comes from a variety of different sources. Writing about Pfam in 2012, Punta et. al [47] note that the data "will come from a variety of sources, in particular, the Protein Data Bank (PDB) and the analysis of complete proteomes for sequences not matched by Pfam." In 2019, speaking of how Pfam has evolved from previous iterations, El-Gebali

---

[3]https://cdla.dev/permissive-1-0/
[4]https://www.ncbi.nlm.nih.gov/genome/browse#!/prokaryotes/

et. al [20] state, "new entries have been deposited in Pfam by the RepeatsDB [a database docused on defining repeats in known structures] curators" and "Evolutionary Classification of Protein Domains (ECOD) ... is a hierarchical classification of protein domains based on evolutionary relationships determined from known structures ... new Pfam entries ... have been generated using ECOD."

- For the training and validation sets from NetSurfP-2.0, Klausen et. al [36] state, "A structural dataset consisting of 12,185 crystal structures was obtained from the Protein Data Bank (PDB)."

- For Protein DataBank, Berman et. al [9] note data is submitted to the database through email or "AutoDep Input Tool (ADIT)" from any person or institution, with the author notified of of a successful submission, or necessary revisions after the ADIT tool annotates for errors.

- For Fluorescence, Sarkisyan et. al [56] state they, "used fluorescence-activated cell sorting and sequenced the entire GFP coding region to assay the fluorescence of many thousands of genotypes created by random mutagenesis of the wildtype sequence."

- For Stability, Rocklin et. al [54] generated the data themselves, by "[expressing oligo library synthesis technology] in yeast so that every cell displays many copies of one protein sequence on its surface... Cells are then incubated with varying concentrations of protease, those displaying resistant proteins are isolated by FACS, and the frequencies of each protein at each protease concentration are determined by deep sequencing (Fig. 1C, for reproducibility of the assay see Fig. S2). We then infer protease $EC_{50}$ values for each sequence from these data by modeling the complete selection procedure (Fig. 1D, details given in Methods)."

- For EuroSat, Helber et. al [29] state that, "The dataset consists of 10 different classes with 2,000 to 3,000 images per class. In total, the dataset has 27,000 images ... [coming from] satellite images taken by the satellite Sentinel-2A ... over 34 European countries."

## C  PII and Offensive Content

To the best of our knowledge, none of the DABS 2.0 datasets contain information directly identifying people involved in the creation of the data. However, satellite imagery may contain enough information to locate the area on Earth the image comes from, and the land or property in the images may be owned by different people or organizations. We are not aware of any offensive content in the DABS 2.0 datasets.

## D  Compute requirements

All runs were performed on an internal cluster with single Titan X GPUs. Most pretraining jobs required between 6 hours to 1 GPU-day, while the transfer jobs ranged from several minutes to approximately 1 GPU-day.

## E  Additional Reproducibility Details

In this section, we describe additional details regarding the processing and use of each dataset. The Genomics, Higgs, and EuroSAT datasets were retrieved using TensorFlow Datasets [1].

### E.1  Bacterial Genomics

The genomic sequences from Genomics OOD [53] are exactly 250 base pairs (one of: A, G, T, C). The sequences are tokenized with a mapping of {A:0, C:1, G:2, T:3}. We use the `train` set for pretraining (split via a 90-10 train-test split). The `validation` and `validation-ood` sets are used for transfer training, and the `test` and `test-ood` sets are used for transfer evaluation.

### E.2  Semiconductor Wafer Manufacturing

Each wafer in WM-811K [74] consists of a 2D array of scalars where 0 represents the background, 1 represents functioning die (semiconducting material), and 2 represents defective die. There is

variation in the size of each wafer. We convert each scalar into a grayscale pigment representation, where 0 is converted to a black RGB pixel representation, 1 is converted to a 50% gray RGB pixel representation, and 2 is converted to a white RGB pixel representation. We resize all the wafers to be 32 x 32, with patch sizes of 4 x 4. The `unlabeled` split is used for pretraining, while the `labeled` split is used for transfer. We construct 90-10 train-test splits for both.

### E.3   Particle Physics

Of the 11M examples in HIGGS [7], we first create a random 90-10 pretrain-transfer split, and then create further 90-10 train-test splits for each. Each 1D scalar tabular feature is mapped to an embedding using a learned affine transformation.

### E.4   Protein Biology

All protein data sequences are tokenized with a given amino acid from "XARNDCQEGHILKMF-PSTWYVUOBZJ" being mapped to its index in the string (the mapping is zero-indexed, meaning for example, the amino acid N is tokenized to 3). All sequences are padded or truncated to 128 tokens, with X being the padding amino acid token. The Pfam dataset is split, leaving 200k of the 31M examples for transfer, and the remainder for pretraining. We produce 90-10 train-test splits for each phase.

### E.5   Multispectral Satellite Imagery

All images from EuroSAT [29] are of size 64 x 64, so there is no need for any resizing. Each channel is standardized to zero mean and unit variance based on the training set statistics. The images are divided into 8 x 8 sized patches before being passed into the embedding layer. We first create a random 90-10 pretrain-transfer split, and then create further 90-10 train-test splits for each phase.