# OpenReview forum: "DABS 2.0: Improved Datasets and Algorithms for Universal Self-Supervision"
_NeurIPS.cc/2022/Track/Datasets_and_Benchmarks — NeurIPS 2022 Datasets and Benchmarks _

### Official Review · Reviewer_1iYW · 2022-07-11
**Good Benchmark**

**Rating:** 6
**Confidence:** 4
**Correctness:** The claims are correct.

**Strengths:**

1. A universal SSL algorithm is proposed along with the benchmark and works as a strong baseline.
2. The insight of evaluating and employing the universal self-supervised on real-world science and engineering datasets would be helpful for both AI for Good and AI for Science.
3. Comprehensive discussion including both internal validity and external validity

**Weaknesses:**

1. The results part mainly explored the masked ratio experiments on the proposed benchmark, compared with two baseline methods. More details on the experiments, models, hyper-parameters are expected to be provided.
2. The documentation of the codebase is constrained into README file without the detailed APIs .

**Additional Feedback:**

1. It would be helpful to see more analysis on the benchmark. For example, for each domain, what is the minimum data requirement to train a strong SSL model. Collecting data of different domains usually has different difficulty.
2. The training details are not very clear. The model used, the parameters, the training time, etc.

**Clarity:**

The paper is well-organized by introducing the dataset and then algorithms. The results and discussions are also provided.

**Documentation:**

Instructions for supporting reproducibility are provided but not very detailed. Datasets details are sufficient.

**Relation To Prior Work:**

Yeah, the paper can be viewed as an extension of DABS 1.0 with proposed SSL algorithm.

**Summary And Contributions:**

The paper proposed a benchmark DABS 2.0, which contains extended real-world science and engineering dataset domains and a universal SSL algorithm.

---

> ### Author Response · Authors · 2022-08-18
> **Response**
>
> We thank the reviewer for their review! We are glad they found our work to be a "good benchmark" which "would be helpful for both AI for Good and AI for Science."
>
> **More details on experiments, models, and hyperparameters**
>
> We thank the reviewer for this suggestion. We have added some more experimental details to Section 4 and Appendix D and E to make the paper more self-contained (including the size of transformer we use, how we process the different modalities, how long we train for, and more details on train/test splits). We initially wanted to avoid cluttering the paper with too many details, but agree that the paper has benefitted from this additional information

---

### Official Review · Reviewer_vbLR · 2022-07-27
**Extension of DABS 1.0 in terms of dataset and universal SSL techniques.**

**Rating:** 6
**Confidence:** 5
**Correctness:** The claims, dataset, algorithms, and …
**Clarity:** The paper is interesting to read and …

**Strengths:**

The paper takes universal SSL one step further in both dimensions: datasets and algorithms. This is a good contribution for the community.

**Weaknesses:**

The 5 new domain datasets are legitimate contributions; however, in my view, MAE does not account as a novel contribution to this research. A lot of implementation details (in section 3) could have been provided to make it a thorough read.

**Additional Feedback:**

-> A typo on line 573. The section is not referred correctly.
-> Line 151 seems to be grammatically incorrect.


**Documentation:**

Yes

**Ethics:**

Everything seems to be intact.

**Relation To Prior Work:**

Yes, but it is an extension to a prior published work.

**Summary And Contributions:**

The authors contribute DABS 2.0, an extension to DABS dataset for universal self-supervision now with additional five new domains. The authors also propose two new algorithms and evaluate them over all the 12 domains with different corruption fractions.

---

> ### Author Response · Authors · 2022-08-18
> **Response**
>
> We thank the reviewer for their review! We are glad they feel our work is "interesting to read" and "is a good contribution to the community"
>
> **"A lot of implementation details could have been provided to make it a thorough read."**
>
> Thank you for this suggestion. We have added some more experimental details to Section 4 and Appendix E to make the section more self-contained (including the size of transformer we use, how we process the different modalities, how long we train for, and more details on train/test splits). We initially wanted to avoid cluttering the paper with too many details, but agree that the paper has benefitted from this additional information.
>
> **"in my view, MAE does not account as a novel contribution to this research"**
>
> We'd like to clarify the scope of our contribution, and what we mean when we write MAE is "a new universal SSL method." A long stream of work has explored the general idea of masked autoencoding across different settings, including Image MAE [1], BERT [2] and before that earlier work such as [3,4] in images. But, typically these MAE is applied in a custom way to each particular application, and typically not explored much outside of these domains.
>
> In this work, we believe we are the first to explore MAE as a *universal* SSL method, applying the same model + training approach across data from 12 different fields. To accomplish this, we do make a few extensions to MAE, including applying it to tabular data (the high-energy physics domain) and multimodal data (the captioned images domain).  However. in the context of the D&B track, we believe our main contribution is the conceptual / benchmarking contribution of establishing MAE as a promising framework for universal SSL. We provide more discussion in Sections 1, 3, and 5 of the updated paper, and are happy to add further clarification.
>
>
> **Additional Feedback**
>
> We appreciate the proofreading suggestions, and have fixed them in the most recent version of the paper.
>
>
> *References*
>
> [1] Masked autoencoders are scalable vision learners, He et al 2022
>
> [2] BERT: Pre-training of Deep Bidirectional Transformers for Language Understanding, Devlin et al 2018
>
> [3] Extracting and composing robust features with denoising autoencoders, Vincent et al 2008
>
> [4] Context Encoders: Feature Learning by Inpainting, Pathak et al 2016

---

### Official Review · Reviewer_6iQg · 2022-07-28
**Extension of a universal self-supervision framework including new datasets and algorithms**

**Rating:** 8
**Confidence:** 2
**Correctness:** Yes.
**Clarity:** Yes.

**Strengths:**

* Extension of a well-established benchmark
* Significant contribution with 5 new datasets domains
* New domains are novel in terms of being less studied than extensively studied, such as images, text, speech, etc...
* Interesting investigation of corruption rate and how it differs based on the domain at hand
* Discussion of external and internal validity



**Weaknesses:**

The scope of the paper is relatively wide with the multiple domains and the two new algorithms. This also makes it difficult to supply a datasheet for all newly introduced datasets, elaborate on the datasets in terms of split, distribution, etc... Additionally, it makes it challenging to cover the details of the benchmarking with a good level of detail.


**Additional Feedback:**

The paper is well-written, I only noticed these minor issues:

* Line 101, an extra “a” is there before “the”
* Line 153, probably a semicolon is better before "namely" for readability
* Line 182, capitalize words of section titles for consistency
* Line 206, probably a simple diagram with inputs (script, domain, algorithm, corruption_frac) is better than having the python command, but I leave this up to the authors.
* Line 266, I think a comma is missing after “However”


**Documentation:**

Yes.

**Ethics:**

Not to my knowledge.

**Relation To Prior Work:**

Yes.

**Summary And Contributions:**

The paper extends the DABS benchmark with datasets from five new domains, proposes a new universal SSL algorithm extending masked auto-encoding. The paper also investigates an interesting metric, the corruption rate across the different domains.

---

> ### Author Response · Authors · 2022-08-18
> **Response**
>
> We thank the reviewer very much for their review! We are glad they appreciate the inclusion of less-studied scientific domains, the investigation of the corruption rate, and the discussion of internal and external validity.
>
> We have added some additional details about each dataset in Appendix E of the most recent version, including details on pretrain/transfer splits, train/test splits, and more details on any preprocessing we do for each dataset (e.g. normalization, tokenization).
>
> We also appreciate the writing suggestions, and have fixed them in the newest version of the paper.

---

### Official Review · Reviewer_16kF · 2022-07-28
**Review for DABS 2.0.**

**Rating:** 7
**Confidence:** 4
**Correctness:** Yes, the paper appears correct.
**Clarity:** Yes, the paper is clear.

**Strengths:**

- Supports generalization of SSL algorithms to less-studied and diverse domains and modalities
- Enables assessment of how certain design choices for one domain affects the performance on another domain. In this work, the cross-domain difference in the influence of masking / permuting of embeddings becomes evident.
- Tasks associated with the datasets introduced are real-world tasks.
- One of the datasets - Bacterial Genomics includes in-distribution and out-of-distribution data enabling an assessment of robustness of models to distributional shifts.
- Each newly introduced dataset is of a different size which can help get an idea of how a domain-agnostic algorithm is able to scale.
- The baseline results indicate that the same SSL algorithm and settings do not perform well in all domains. This is the challenge for SSL algorithms - to generalize to multiple domains without needing manual domain-specific tuning. The fact that the results indicate this challenge, supports the purpose of this dataset.


**Weaknesses:**

- Easy execution of pre-training and transfer learning with the help of newly added code is mentioned but the code / dataset are not made available for review.
- Claims surrounding 'Universal SSL' may be a bit too broad, since graphs, point clouds, etc. are not included.

**Additional Feedback:**

_Minor Errors:_
- The DABS 1.0 image in Fig. 1 can be a bit bigger.

_Notes to the author:_
-  Performance could be more easily interpreted if the performance data in the table were presented in a different format (plots with performance as a function of corruption level)



**Documentation:**

Easy execution of pre-training and transfer learning with the help of newly added code is mentioned but the code / dataset are not made available for review. (As noted in weaknesses)

**Ethics:**

No concerns noted.

**Relation To Prior Work:**

Yes, the authors discuss prior work.

**Summary And Contributions:**

Five new real-world datasets in science and engineering are added to the original 7 datasets in DABS 1.0. Also, two additional unsupervised learning methods are introduced - Capri, and a generalized masked encoding. Both these algorithms and the ShEd algorithm from DABS 1.0 are then applied to all the 12 datasets and the variation of influence of corruption fraction on performance, across the 12 domains (and the three algorithms) is observed. The process and results convey the potential of using DABS 2.0 for assessing robustness of domain-agnostic SSL methods and also for studying the effect of design decisions across domains.

---

> ### Author Response · Authors · 2022-08-18
> **Response**
>
> We thank the reviewer very much for their review! We are glad they appreciate the broader motivation behind the work, in addition to the methods and results.
>
> The suggestions for the figure and table are helpful and appreciated. We have increased the size of the DABS 1.0 image, and will experiment more with visualizing the table as a plot for our next revision.
>
> We are still in the process of cleaning and commenting the code, but it will be made public as part of the DABS repo (https://github.com/alextamkin/dabs) as soon as possible, incorporating the datasets and algorithms in the same way as in DABS 1.0. For example, to pretrain MAE on the proteins dataset, one will be able to run `pretrain.py dataset=proteins algorithm=mae`.

---

### Official Review · Reviewer_WxUa · 2022-07-29
**Review for DABS 2.0: Improved Datasets and Algorithms for Universal Self-Supervision**

**Rating:** 4
**Confidence:** 3
**Clarity:** The paper is reasonably well written.

**Strengths:**

The authors consider an interesting set of datasets from a wide range of disciplines. Given the wide range of datasets covered the updated DABS benchmark should be of interest to the broad community. To the best of my understanding, the authors have properly answered questions regarding the datasets containing offensive content.

**Weaknesses:**

I have several concerns regarding the paper.

Significance of the contribution:

1. Out of the five datasets that the authors use, three of them (higgs, genomics, eurostat) are directly available in tensorflow already. This implies the datasets are already in a usable format and I do not see what is the authors contribution in terms of these datasets.  The introduction and description in the main text feels misleading.

2. The two algorithms that the authors use are a minor variation of MAEs and contrastive learning. Therefore, I do not think there is any insightful algorithmic variation that the authors introduce either. I understand that this is the benchmarks track and we do not expect completely new methods. Despite that I think what is introduced should not be stated as a new universal SSL method but should be honestly acknowledged as natural extensions/variations of well known approaches.

3. Thirdly, the authors show a table of results. I did not gather any new insights from the results and that was not very pleasing.

4. Finally, there is a bunch of experiments that the authors state ran into an issue and have been marked as pending (for some other results are marked OM). I understand that issues can happen and do not want to penalize authors for this. At the same time, it seems unfair to others. By accepting this we are allowing authors extra time to gather results that others did not have.

**Additional Feedback:**

I do not have additional feedback. I look forward to authors responses as I might have missed something.

**Correctness:**

In the main table, I am guessing authors are reporting accuracy? Also, what is going on with OM and DV?

**Documentation:**

The URL that authors give seems to contain only information about DABS and not DABS 2.0. However, the authors state that they will release the code. Therefore, I cannot evaluate reproducibility at this point.

**Ethics:**

To the best of my knowledge, there are no ethical concerns.

**Relation To Prior Work:**

Yes, the authors have discussed prior work. However, i feel the contributions on top of DABS benchmark are limited and do not seem to merit a full paper yet.

**Summary And Contributions:**

In this work, the authors extend the DABS benchmark [1] to include five more datasets. The new datasets are carefully chosen to cover a variety of underserved domains -- i) bacterial genomics, ii) semiconductor wafer manufacturing, iii) particle physics, iv) protein biology, v) satellite imagery. The authors provide evaluation of two new variations of masked autoencoders and contrastive learning under varying levels of corruption (masking) in the data for the five new datasets and the seven datasets from the original DABS benchmark.  Generalized MAE does well on the satellite image dataset but it is not consistently the case across domains. Furthermore, the optimal amount of corruption is not just domain dependent and can depend on other factors making it hard to choose.

[1] Tamkin, Alex, et al. "Dabs: A domain-agnostic benchmark for self-supervised learning." Neurips 2021 datasets and benchmarks track.

---

> ### Author Response · Authors · 2022-08-18
> **Response (1/2)**
>
> We thank the reviewer very much for their review! We are happy they found our work to be an "interesting set of datasets from a wide range of disciplines" and that our work "should be of interest to the broad community"
>
> We have updated the manuscript reflecting the below discussion. We hope these responses and revisions answer the most important questions, and we are very happy to engage further throughout the remainder of the discussion period.
>
> **1. Three datasets are "available in tensorflow already. This implies the datasets are already in a usable format and I do not see what is the authors contribution in terms of these datasets. The introduction and description in the main text feels misleading."**
>
> We'd like to emphasize that our primary contribution is not making these datasets easy to use, but in
> 1. identifying neglected STEM datasets that are promising for SSL (Section 2)
> 2. adapting these datasets for general self-supervised learning (Sections 4 and 5), and
> 3. providing a detailed discussion motivating real-world STEM datasets as a promising direction for ML benchmarking (Section 6).
>
> Some of the most influential benchmarks, including GLUE [1] and SuperGLUE [2], derived their value in a similar way, via well-motivated selection of existing datasets to drive and measure progress. We are a bit confused about what may have appeared misleading, as we do cite the sources of these datasets in the main text (Section 2), and don't claim to have collected them ourselves, similar to the original DABS paper. That said, we are very happy to make changes here if it would add further clarity.
>
> In addition, even in the cases where we access several raw datasets via TensorFlow, we did do further work to make these datasets usable in a benchmark like DABS, including:
> - Casting each dataset as a pretrain/transfer problem (e.g. deciding which datasets should be used for pretraining vs transfer, and how to split the data to accomplish this)
> - Within each phase (pretrain / transfer) deciding train/test splits, which are not always present (e.g. for EuroSAT and HIGGS)
> - Standardizing preprocessing for each dataset, including normalization (e.g. standardizing spectral bands for satellite imagery) and tokenization / numericalization (e.g. breaking apart genomic sequences into nucleotide indices)
>
> **2. "The two algorithms that the authors use are a minor variation of MAEs and contrastive learning…I think what is introduced should not be stated as a new universal SSL method but should be honestly acknowledged as natural extensions/variations of well known approaches"**
>
> We'd like to clarify the scope of our contribution, and what we mean by "a new universal SSL method." A long stream of work has explored the general idea of masked autoencoding across different settings, including Image MAE [3], BERT [4] and before that earlier work such as [5,6] in images. In the same way, similar approaches to Capri were explored in a limited way on image [7] and audio data [8]. But, typically these algorithms are applied in a custom way to each particular application, and typically not explored much outside of two domains.
>
> In this work, we believe we are the first to explore MAE and Capri as *universal* SSL methods, applying the same model + training approach across data from 12 different fields. To accomplish this, we do make a few extensions to MAE, including applying it to tabular data (the high-energy physics domain) and multimodal data (the captioned images domain). However. in the context of the D&B track, we believe our main contribution is the conceptual / benchmarking contribution of establishing MAE as a promising framework for universal SSL. We provide more discussion in Sections 1, 3, and 5 of the updated paper, and are happy to add further clarification.

---

> > ### Author Response · Authors · 2022-08-18
> > **Response (2/2)**
> >
> > **3. Experiments marked with pending, OM, or DV**
> >
> > We're happy to share the updated version of our table in the most recent version, including the pending runs. Following the DABS paper, we report the average validation metrics across the domain (generally accuracy but also AUROC, pearson/spearman correlation, or F1). Here are additional details for OM and DV, which we have added to the most recent revision (L227-231):
> > - OM: Out of Memory. This denotes runs that resulted in an Out of Memory error, due to the expensive softmax computation for large vocabulary sizes. This is especially the case for 85% masking (which results in more softmax computations) and in the multilingual text datasets, which have a larger vocabulary size (resulting in a more expensive computation per softmax). These results reflect one tradeoff of MAE as an approach: namely, that the objective is less general than ShED or Capri, leading to uneven compute requirements across datasets.
> > - DV: Diverged. The contrastive Capri objective is at times unstable on certain datasets, resulting in runs where the loss diverges to NaN. These models cannot not be transferred to the downstream datasets, and reflect a tradeoff of Capri: while it has more consistent compute requirements vs MAE, it also can fail on certain datasets.
> >
> >
> > **4. Clarifying new insights from the results**
> >
> > Thank you for the suggestion. We have added a summary of takeaways from the results (see **Takeaways**, Section 5), which we believe much improves this section. These include:
> > - ShED and especially MAE routinely show notable and consistent gains over Scratch, showing their potential as general approaches
> > - However, the best algorithm varies by domain, as does the best masking fraction, meaning that while both ShED and MAE are often good choices, there is still no "single best algorithm." This highlights an important direction for future research.
> > - Moreover, there are some tradeoffs between algorithms—MAE outperforms ShED on several domains, but can be too memory expensive to run on tokenized datasets with large vocabulary sizes
> > - Capri still shows some gains over Scratch, but performs less well overall and sometimes diverges on certain datasets
> >
> > These results help us understand the capabilities and limitations of existing general SSL methods, and give low-resource domain experts a useful set of tools and starting points they can apply to their own problems.
> >
> >
> > *References*
> >
> > [1] GLUE: A multi-task benchmark and analysis platform for natural language understanding, Wang et al, 2018
> >
> > [2] Superglue: A stickier benchmark for general-purpose language understanding systems, Wang et al, 2019
> >
> > [3] Masked autoencoders are scalable vision learners, He et al 2022
> >
> > [4] BERT: Pre-training of Deep Bidirectional Transformers for Language Understanding, Devlin et al 2018
> >
> > [5] Extracting and composing robust features with denoising autoencoders, Vincent et al 2008
> >
> > [6] Context Encoders: Feature Learning by Inpainting, Pathak et al 2016
> >
> > [7] Selfie: Self-supervised Pretraining for Image Embedding, Trinh et al 2019
> >
> > [8] MERLOT Reserve: Neural Script Knowledge through Vision and Language and Sound, Zellers et al 2022

---

### Meta-Review · Area_Chair_mnRk · 2022-09-09

**Recommendation:** Accept
**Confidence:** 5

**Metareview:**

DABS 2.0 extension of DABS to include five more datasets, and serves as benchmark for self-supervised representation learning. The new datasets cover new domains such as genomics, industrial images, biology, satellite imagery, etc. In addition, two new self-supervised learning methods are evaluated and their robustness to domains and hyper parameters are evaluated. In addition a new technique called Capri is introduced, which combines benefits of masked auto-encoders and contrastive learning to learn representations.

The reviewers are generally positive except R1 who recommends reject. The main concern of R1 is that the contributions over the DABS is not significant (e.g., several datasets are available in tensor flow). Contrast this with R2, R3, R4, R5 who find the contributions valuable. Given the importance of unsupervised representation learning and the significant effort being put in the research community this work is valuable. Thus they recommend accept.

---

### Decision · Program_Chairs · 2022-09-16

Accept